# Analysis of ancestry-specific polygenic risk score and diet composition in type 2 diabetes

**Dale S. Hardy**[1]*, **Jane T. Garvin**[2], **Tesfaye B. Mersha**[3]

**1** Department of Internal Medicine, Morehouse School of Medicine, Atlanta, GA, United States of America, **2** College of Nursing, Walden University, Minneapolis, MN, United States of America, **3** Cincinnati Children's Hospital Medical Center, University of Cincinnati College of Medicine, Cincinnati, OH, United States of America

☉ These authors contributed equally to this work.
* dhardy@msm.edu

**Data Availability Statement:** Datasets utilized in our study are publicly available from dbGaP (https://www.ncbi.nlm.nih.gov/gap/) as follows: ARIC (phg000035, phs000280); CARDIA (phg000098, phs000285); CHS (phg000077,

## Abstract

### Background

Carbohydrate and protein dietary proportions have been debated as to whether higher or lower levels are optimal for diabetes metabolic control.

### Objective

The objective of this study was to investigate the associations, interactions, and mediational relationships between a polygenic risk score (PRS), carbohydrate and protein intake, and physical activity level on type 2 diabetes (T2DM) by genetic ancestry, in European Americans and African Americans. A secondary objective examined the biological pathways associated with the PRS-linked genes and their relationships to dietary intake.

### Methods

We performed a cross-sectional study in 9,393 participants: 83.3% European Americans and 16.7% African Americans from 7-NHLBI Care studies obtained from the database of Genotypes and Phenotypes. The main outcome was T2DM. Carbohydrate and protein intake derived from food frequency questionnaires were calculated as percent calories. Data were analyzed using multivariable generalized estimation equation models to derive odds ratios (OR) and 95% confidence intervals (CI). Ancestry-specific PRSs were constructed using joint-effects Summary Best Linear Unbiased Estimation in the train dataset and replicated in the test dataset. Mediation analysis was performed using VanderWeele's method.

### Results

The PRS in the highest tertile was associated with higher risk of T2DM in European Americans (OR = 1.25; CI = 1.03–1.51) and African Americans (OR = 1.54; 1.14–2.09). High carbohydrate and low protein intake had lower risks of T2DM when combined with the PRS after adjusting for covariates. In African Americans, high physical activity combined with the high PRS and high protein diet was associated with a 28% lower incidence of T2DM when

phs000287); FHS OFFSPRING (phg000006, phs000007); FHS GENX3 (phg000006, phs000007); MESA (phs000071, phs000209); WHI (phg0000398, phg0000148, phg000061, phs000200).

**Funding:** This work is supported by a K01 grant from the National Heart, Lung, and Blood Institute (NHLBI) grants: K01 HL127278 awarded to Dale Hardy, PhD. There was no additional external funding received for this study. The NHLBI had no role in study design, data collection and analysis, decision to publish, or preparation of the manuscript.

**Competing interests:** The authors have declared that no competing interests exist.

compared to low physical activity. In mediational models in African Americans, the PRS-T2DM association was mediated by protein intake in the highest tertile by 55%. The top PRS tertile had the highest magnitude of risks with metabolic factors that were significantly associated with T2DM, especially in European Americans. We found metabolic pathways associated with the PRS-linked genes that were related to insulin/IGF and ketogenesis/keto-lysis that can be activated by moderate physical activity and intermittent fasting for better T2DM control.

## Conclusions

Clinicians may want to consider diets with a higher portion of carbohydrates than protein, especially when the burden of high-risk alleles is great in patients with T2DM. In addition, clinicians and other medical professionals may want to emphasize the addition of physical activity as part of treatment regimen especially for African Americans. Given the metabolic pathways we identified, moderate physical activity and intermittent fasting should be explored. Researchers may want to consider longitudinal or randomized clinical trials to determine the predictive ability of different dietary patterns to inhibit T2DM in the presence of obesity and an elevated PRS.

## Introduction

Type 2 diabetes (T2DM) is one of the fastest growing chronic non-communicable disease conditions today that affects 37.3M (11.3%) people in the United States of America (USA) [1]. It is estimated that the age-adjusted prevalence of T2DM is 12.7% in African Americans vs. 7.4% in European Americans, aged 18 years or older [2]. A distressing fact is that about 1 in 5 adults with T2DM and 8 in 10 adults with pre-diabetes are undiagnosed [1]. Furthermore, by the time patients are diagnosed, because of the destructive insidious nature of T2DM, a sizeable proportion of the population already have diabetic complications such as neuropathy and retinopathy, implicated from chronically high glucose levels and high blood pressure [1, 3]. T2DM, a major cause of heart disease and stroke, is the 7th leading cause of death in the USA [1]. Several risk factors are associated with T2DM and its complications, including genetics, imbalanced diets, impaired glucose metabolism, advanced age, physical inactivity, and genetic ancestry [1, 4].

Genome-Wide Association Studies (GWAS) have discovered multiple variants with small effects associated with common disorders, such as T2DM [5]. Due to smaller effects from each variant, many studies aggregate the effects of multiple single nucleotide polymorphisms (SNPs) to create a polygenic risk score (PRS) to predict common disorders, such as T2DM [5]. PRSs have been constructed primarily in populations of European ancestry. Liu et al. [6] developed a PRS with a tuning parameter that had the highest area under the curve statistic that improved the predictive ability for T2DM risk. A hypertension PRS has been associated withT2DM, as well as coronary artery disease, ischemic stroke, and kidney disease [7]. Furthermore, a Multi-PRS from 81 GWAS summary statistics that included the joint effect of interaction of SNPs, predicted 5.45% of BMI as well as other socio-environmental traits [8].

Healthy diets and higher physical activity levels are associated with adequate metabolic control of T2DM [1]. However, carbohydrate and protein dietary proportions have been largely debated as to whether high or low levels are optimal for diabetes metabolic control. Currently,

the American Diabetes Association [9] does not recommend individualized diets with specific carbohydrate or protein proportions for T2DM control. Our published meta-analysis on carbohydrate quality found that high-glycemic carbohydrate diets were associated with higher rates of developing T2DM [10]. Moreover, the carbohydrate quality represented rapidly available carbohydrates that increased T2DM risk. Another systematic review found that there was no association between a low carbohydrate diet score and T2DM risk [11]. But the authors found that carbohydrate intake within the recommended 45% to 65% of total calorie intake was not associated with an increased risk of T2DM; yet carbohydrate intake > 70% of calorie intake might be associated with a higher risk. However, another study reported that there was no interaction between a BMI associated PRS and caloric intake [12].

Moderate to high physical activity, performed 120 to 150 minutes per week, has been associated with reduced risk of developing T2DM, metabolic syndrome, cardiovascular disease, and stroke [1]. Furthermore, over 20-year time periods, lower levels of physical activity have been associated with elevated T2DM and metabolic risk factors [13] but a higher fitness level was associated with lower risk for developing incident prediabetes/T2DM [14]. Because T2DM is a highly polygenic disorder that may differ by genetic ancestry, understanding the biological mechanism linking genes with related T2DM SNPs may give further insight into the pathways involved. Our main study aim was to investigate the associations, interactions, and mediational relationships between the PRS, carbohydrate and protein intake, and physical activity level by genetic ancestry to increase T2DM risk. A secondary aim examined the biological pathways associated with the PRS-linked genes and their relationship to dietary intake.

## Materials and methods

Participants were drawn from seven National Heart, Lung, and Blood Institute (NHLBI) Candidate Gene Association Resource (Care) studies that were obtained from the database of Genotypes and Phenotypes (dbGaP) [15]. We combined de-identified data from the Atherosclerosis Risk in Communities (ARIC) study [16], Coronary Artery Risk Development in Young Adults Study (CARDIA) [17], Cardiovascular Heart Study, (CHS) [18] Framingham Heart Study (FHS) Offspring and FHS GENX 3 studies [19], Multi-Ethnic Study of Atherosclerosis Study (MESA) [20], and Women's Health Initiative study (WHI) [21] to create a combined dataset by genetic ancestry (S1 Table). All studies were sponsored by the NHLBI, and are large-scale, ongoing prospective cohort studies focused on atherosclerosis conditions and their sequelae. We previously intended to use Jackson Heart Study (JHS) [22], but a high proportion of the sample were already in ARIC; and after merging genetic datasets, administering quality control procedures, and applying filtering criteria, these observations did not make it in the combined dataset. Further design and sampling are explained elsewhere for all included studies [16–21, 23]. All participants signed an informed consent as part of the initial study. Our subsequent study was approved by the Social & Behavioral Institutional Review Board at Morehouse School of Medicine.

### Data harmonization

We harmonized our data by bringing together data of varying formats, such as file formats, variable definitions, etc. from the seven NHLBI Care datasets in order to generate a large cohesive dataset (S2 Table). We assessed all chosen variables to ensure their presence in all NHLBI Care datasets. Some variables were transformed to yes/no status to harmonize measures across datasets. For example, the physical activity variable had different formats and meaning across datasets. In order to harmonize the physical activity variable across datasets, we recoded this variable as a high/low form by evaluating its functional form in each dataset. When we selected

study variables that were present in all 7-NHLBI Care datasets, we created variable definitions using existing variable definitions that agree across all datasets, so our chosen variables had consistent meaning across all datasets.

## Food frequency questionnaire assessment

Studies that used a semi-quantitative food frequency questionnaire (FFQ) to obtain information on dietary intake were ARIC [24] and FHS studies [25], and MESA [26]. Studies that used diet history recalls were CARDIA [27] and CHS [28]. S3 Table shows more details of these studies. Generally, in the FFQ responses, participants reported their intake based on 9-levels of frequency, ranging from < 1 time per month to ≥ 6 times per day. During the diet history sessions, participants were asked questions about usual intake. At the examination interview, participants were shown standard serving sizes and typical servings using food models and were asked brand names of prepared foods to help them estimate intake.

## Study variables

We defined T2DM, the outcome variable according to the American Diabetes Association criteria [29], by one or more the following: fasting blood glucose ≥ 126mg/dL, non-fasting blood glucose ≥ 200 mg/dL, self-reported diabetes diagnosis, or taking diabetes medications. T2DM status and all covariates were measured at baseline except for data from FHS Offspring and GENX 3 which were taken from Exam 3 and Exam 2 respectively. In addition, MESA baseline data were taken from Exam 5 because Exam 1 was not available in dbGaP data. Covariates considered for adjustment were age (dichotomized), sex, physical activity (low/high), current smoking (yes/no), current drinking (yes/no), total calories (dichotomized), percent carbohydrate or percent protein intake in tertiles, percent total fat intake in tertiles, and 10 principal components to correct for population stratification.

## Statistical analysis

The PRS and 10 principal components to correct for population stratification (n = 12,923) were calculated from the test dataset, taken from a random 50% of the data. The other random 50% of the data was used as the train set. We excluded participants if they had missing observations at baseline for T2DM (n = 2,595), dietary carbohydrate and protein intake, and total caloric intake (n = 776), total caloric intake < 600 Kcal or > 4,200 Kcal per day for men (n = 85), and < 500 Kcal or > 3,600 Kcal per day for women (n = 74). Our final models at baseline included 9,393 participants in which 7,822 (83.3%) were European Americans and 1,571 (16.7%) were African Americans. We imputed missing observations on current smoking status, current drinking status, and physical activity level to augment our sample, especially for African Americans. We performed multiple imputations using predictive mean matching for continuous variables and logistic regression for binary variables. We specified 50 imputations per imputed variable, then specified mean values which were used in our analysis Imputations were < 5% of the original participant sample.

We computed a covariate propensity summary score by genetic ancestry to decrease bias by regressing T2DM on the covariates (age, sex, physical activity, cigarette smoking status, current drinking status, total calories, % carbohydrate or % protein intake, % total fat intake, and 10 principal components for population stratification). Our data were analyzed using generalized estimation equation (GEE) models to derive odds ratios (OR) and 95% confidence intervals (CI). We applied within-subject identification to specify measured and unmeasured correlations nested in the participant using the independent correlational structure to boost the model efficiency. In our GEE analysis, we were interested in the expectation of the

outcome, T2DM, as a function of the PRS and/or carbohydrate or protein intake adjusted for the covariate propensity score.

In our GEE multivariable models, we regressed T2DM on the PRS and carbohydrate or protein intake, adjusting for the covariate propensity score. In the mediation models, we included the PRS, % carbohydrate or % protein intake, along with the covariate propensity score for adjustment. We evaluated the effects of high physical activity level in combination with the functional risk-raising PRS and high protein intake vs. participants with low physical activity level using the risk difference (pg. 47) [30].

We were interested in causal mediation using the counterfactual approach to assess relationships [31]. All analyses were performed in tertiles of carbohydrate or protein intake by PRS tertiles. In all models, a 2-sided $p < .05$ and more stringent, Bonferroni adjustment for multiple testing ($p < .0125$) were used as the threshold for statistical significance. All regression analyses were bootstrapped 10,000 times. Our multivariable statistical analyses were conducted using Stata MP, version 16.0 (StataCorp, College Station, TX).

## Constructing the PRS and principal components for stratification

We performed SNP imputation using the Michigan Imputation Server algorithm utilizing 1000 Genomes Phase 3 (Version 5) to increase the number of SNP markers for our PRS construction and statistical power to detect associations in our analysis [32]. After quality cleaning and merging imputed datasets, SNPs for the PRS were extracted using Plink, a whole genome association analysis toolset [33] by genetic ancestry. For SNPs within high linkage disequilibrium $\geq .8$, tag SNPs were chosen based on higher binding capacity in RegulomeDB [34]. The Hardy-Weinberg test for all SNPs was performed in Plink [33] using chi-square goodness-of-fit test for European Americans and African Americans separately. The 10 genetic principal components were computed in Linux using Genome-wide Complex Trait Analysis 64 (GCTA64) guidelines [35] to calculate a genetic-related matrix by genetic ancestry and then specifying the 10 principal components.

## DNA features and regulatory regions

We used clumping and threshold methodology tuning parameters for the PRS. After we clumped the SNPs and performed lasso shrinkage, we regressed T2DM against each SNP by genetic ancestry (European American or African American). SNPs with a $p < .05$ were chosen for further analysis. We flipped alleles for SNPs that had lower T2DM risks, making their effects risk-raising. To choose causal SNPs associated with T2DM, we selected SNPs with a RegulomeDB score of $\leq 4$ that contained DNA features and regulatory regions for at least transcription factor binding sites together with DNase peaks or motifs to include in our working genetic dataset [34]. RegulomeDB presents a scoring system with functional categories ranging from 1 to 6 by the way of integrated annotations data on methylation, chromatin structure, protein motifs and binding. The lower the RegulomeDB score, the stronger the evidence for a variant to be in a functional genomic region. The PRS was trained using phenotypes for glucose and T2DM. To increase the predictive power of the PRS, the joint power of multiple SNPs was included in the PRS using the summary best linear unbiased prediction (SBLUP) method. We then generated summary statistics on the train dataset and replicated these statistics in the test dataset to create our PRS to use in our statistical models.

## Results

### Descriptive characteristics by demography

In this cross-sectional study, there were 9,393 participants: 7,822 (83.3%) European Americans and 1,571 (16.7%) African Americans. Table 1 shows the descriptive characteristics of the sample by genetic ancestry. All characteristics except age were significantly different between European Americans and African Americans. Compared to European Americans, African Americans had a higher average BMI and a doubling of obesity and T2DM rates and higher blood glucose levels; were less physically active; tended to be current smokers; and a greater percentage had elevated systolic blood pressure and were on blood pressure medications. A higher proportion of European Americans were at normal weight, were more physically active, and had higher HDL levels; but a higher proportion drank alcohol and had elevated triglyceride levels.

### Association of the PRS by genetic ancestry

We constructed PRSs by genetic ancestry in Europeans Americans and African Americans separately. After merging the seven NHLBI Care imputed datasets, we obtained 5,957,358 markers each in European Americans and African Americans. After quality control procedures such as pruning, cleaning, and within Hardy-Weinberg equilibrium, there remained 120,991 variants in European Americans and 265,042 variants in African Americans. After

**Table 1. Characteristics among European Americans and African Americans at baseline.**

| | Both Racial Ancestries Combined | European Americans | African Americans | P Value Comparing African Americans to European Americans |
|---|---|---|---|---|
| | (n = 9,393) | (n = 7,822) | (n = 1,571) | |
| **Characteristic** | **Mean (SD) or Column Percent (%) of Participants** | | | |
| Age (20–94 years) | | | | .093 |
| 20–57 years | 48.1 (7.4) | 47.5 (7.8) | 50.5 (4.1) | |
| 58–94 years | 67.6 (7.7) | 67.9 (7.7) | 66.2 (7.7) | |
| Body mass index (BMI) | 27.3 (5.2) | 26.8 (4.8) | 29.8 (6.1) | < .0001 |
| Weight status (%) | | | | < .0001 |
| Underweight (BMI $\leq$ 18.5 kg/m$^2$) | 1.2 | 1.2 | 1.3 | |
| Normal weight (BMI 18.5–24.9 kg/m$^2$) | 33.9 | 37.3 | 16.3 | |
| Overweight (BMI 25.0–29.9 kg/m$^2$) | 39.5 | 40.3 | 35.8 | |
| Obese (BMI $\geq$ 30 kg/m$^2$) | 25.4 | 21.3 | 46.6 | |
| High physical activity level (%) | 54.8 | 56.2 | 47.4 | < .0001 |
| Current cigarette smoking (%) | 30.8 | 30.2 | 33.5 | .010 |
| Current alcohol intake (%) | 50.7 | 53.2 | 37.4 | < .0001 |
| Systolic blood pressure (% > 120mmHg) | 53.3 | 51.0 | 65.2 | < .0001 |
| HDL (% < 40mg/dL: men and < 50mg/dL: women) | 34.9 | 35.6 | 30.8 | < .0001 |
| Triglycerides (% >150mg/dL) | 27.5 | 29.4 | 17.7 | < .0001 |
| Fasting blood glucose (% > 100 mg/dL) | 45.6 | 44.4 | 52.0 | < .0001 |
| Blood pressure medications (%) | 32.4 | 28.8 | 51.3 | < .0001 |
| Type 2 diabetes (%) | 14.1 | 12.2 | 24.3 | < .0001 |

**Abbreviations:** HDL, high density lipoprotein cholesterol; sd, standard deviation

%, percent of sample. All other variable results are means.

P values for proportions of categorical variables among European Americans and African Americans were calculated using Pearson's chi-square tests of hypothesis for independence. Analysis showed that all variables were statistically significant between each other.

**Table 2. Association between a polygenic risk score (PRS) and type 2 diabetes at baseline.**

| | Odds Ratio(95% Confidence Interval) P Value | | | |
| --- | --- | --- | --- | --- |
| | European Americans | P Value | African Americans | P Value |
| | (n = 7,822) | | (n = 1,571) | |
| **Polygenic risk score** | 216 SNPs | | 30 SNPs | |
| PRS: per 1 SD increase in Z score | **1.10 (1.02–1.19)** | **.012*** | **1.16 (1.02–1.31)** | **.023*** |
| PRS lowest tertile | 1.00(ref) | | 1.00(ref) | |
| PRS second tertile | 1.12 (0.93–1.36) | .239 | 1.26 (0.93–1.73) | .141 |
| PRS highest tertile | **1.25 (1.03–1.51)** | **.021*** | **1.54 (1.14–2.09)** | **.005*** |

**Abbreviation:** SNPs, single nucleotide polymorphisms.

Bold indicates p values that were statistically significant at p < .05.

*Bonferroni adjustment for multiple testing (p < .05/2 = < .025).

Type 2 diabetes status was regressed against each macronutrient using Generalized Estimating Equation analysis, adjusting for a covariate propensity score composed of age, sex, physical activity, current cigarette smoking status, current drinking status, and 10 principal components for population stratification to derive odds ratios and 95% confidence intervals.

further administering clumping and lasso shrinkage techniques, our datasets diminished to 3,884 variants in European Americans (n = 24,908), and 2,177 variants in African Americans (n = 4,036). The genotyping rates were 99.99% in European Americans and 100% in African Americans.

The PRS was associated with T2DM and had higher magnitude of associations in African Americans (Table 2). Each unit in Z-score per 1-SD increase in the PRS was associated with a 10% higher T2DM risk in European Americans (Odds Ratio (OR) = 1.10; 95% Confidence Interval: 1.02–1.19) and a 16% higher risk (OR = 1.16;1.02–1.31) in African Americans. We found higher risk for T2DM when we compared the higher PRS tertile to the lowest tertile. For European Americans, we found a 25% higher T2DM risk (OR = 1.25; 1.03–1.51) and 54% higher T2DM risk for African Americans (OR = 1.54; 1.14–2.09). All effect estimates mentioned passed Bonferroni cut-off for false discovery rate at p < .0125.

## Association of carbohydrate and protein intake with T2DM

Table 3 shows the association of carbohydrate and protein intake on T2DM. High carbohydrate and low protein intake had lower risk in their association with T2DM. For European Americans, there were statistically significant T2DM risks with carbohydrate intake in the highest tertile compared to the lowest tertile (OR = 0.77;0.64–0.92); and for protein intake in the second (OR = 0.79; 0.66–0.94) and lowest (OR = 0.65; 0.54–0.78) tertiles compared to the highest tertile. For African Americans, we only saw statistically significant T2DM risk for protein intake in the lowest tertile compared to the highest tertile (OR = 0.61; 0.46–0.81). However, when we examined the reverse for diets low in carbohydrate and high in protein, this resulted in higher risks in their association with T2DM. The results for European Americans revealed that the lower tertile of carbohydrate (OR = 1.30; 1.09–1.57) and the highest tertile of protein intake (OR = 1.54; 1.28–1.84) were associated with higher risks of T2DM. The risks for African Americans were fewer but were higher in magnitude than risks for European Americans. For African Americans, we only observed higher T2DM risk in the highest tertile vs. lowest tertile of protein (OR = 1.65; 1.24–2.20) that met the Bonferroni cut off of p < .0125.

Physical activity had a decreased risk with T2DM in both ancestries. European Americans showed 54% decreased T2DM risk (OR = 0.46; 0.39–0.55); and African Americans had 34% lower T2DM risk (OR = 0.66; 0.51–0.85).

**Table 3. Association between different combinations of carbohydrate and protein intake, and physical activity alone on type 2 diabetes.**

| | Odds Ratio(95% Confidence Interval) P Value | | | | |
|---|---|---|---|---|---|
| | European Americans | P Value | African Americans | | P Value |
| | (n = 7,822) | | (n = 1,571) | | |
| **Carbohydrate (high vs. low)** | | | **Carbohydrate (high vs. low)** | | |
| Lowest tertile (126g/39%) | 1.00(ref) | | Lowest tertile (122g/38%) | 1.00(ref) | |
| Second tertile (195g/48%) | 0.95(0.80–1.13) | .578 | Second tertile (194g/48%) | 0.88(0.65–1.18) | .389 |
| Highest tertile (301g/58%) | **0.77(0.64–0.92)** | **.004\*** | Highest tertile (308g/59%) | 0.81(0.60–1.08) | .149 |
| **Protein (low vs. high)** | | | **Protein (low vs. high)** | | |
| Highest tertile (109g/22%) | 1.00(ref) | | Highest tertile (106g/22%) | 1.00(ref) | |
| Second tertile (71g/17%) | **0.79(0.66–0.94)** | **.008\*** | Second tertile (71g/17%) | 0.79(0.59–1.06) | .114 |
| Lowest tertile (46g/14%) | **0.65(0.54–0.78)** | **.001\*** | Lowest tertile (44g/13%) | **0.61(0.46–0.81)** | **.001\*** |
| **Reverse of Above** | | | **Reverse of Above** | | |
| **Carbohydrate (low vs. high)** | | | **Carbohydrate (low vs. high)** | | |
| Highest tertile (301g/58%) | 1.00(ref) | | Highest tertile (308g/59%) | 1.00(ref) | |
| Second tertile (195g/48%) | **1.24(1.03–1.49)** | **.020** | Second tertile (194g/48%) | 1.09(0.82–1.45) | .568 |
| Lowest tertile (126g/39%) | **1.30(1.09–1.57)** | **.004\*** | Lowest tertile (122g/38%) | 1.24(0.93–1.66) | .149 |
| **Protein (high vs. low)** | | | **Protein (high vs. low)** | | |
| Lowest tertile (46g/14%) | 1.00(ref) | | Lowest tertile (44g/13%) | 1.00(ref) | |
| Second tertile (71g/17%) | **1.21(1.01–1.46)** | **.035** | Second tertile (71g/17%) | 1.30(0.96–1.77) | .087 |
| Highest tertile (109g/22%) | **1.54(1.28–1.84)** | **.001\*** | Highest tertile (106g/22%) | **1.65(1.24–2.20)** | **.001\*** |
| Physical activity (high vs. low) | **0.46(0.39–0.55)** | **< .0001\*** | Physical activity (high vs. low) | **0.66(0.51–0.85)** | **.001\*** |

Bold indicates p values that were statistically significant at p < .05. *Bonferroni adjustment for multiple testing (p < .05/4 = < .0125).

Macronutrient intakes were calculated as percent of total caloric intake. Type 2 diabetes status was regressed against each macronutrient using Generalized Estimating Equation analysis, adjusting for a covariate propensity score composed of age, sex, physical activity, current cigarette smoking status, current drinking status, carbohydrate, protein, and fat intake to derive odds ratios and 95% confidence intervals. When a specific nutrient was the focus of interest, it was not included in the covariate propensity score.

## Association of carbohydrate and protein intake with physical activity on T2DM

We next examined the association for carbohydrate and protein intake in the presence of physical activity. In Figs 1 and 2, we observed that the combination of a high carbohydrate or low protein intake with high physical activity level was associated with lower T2DM risks than their separate parts (See Table 3). In European Americans (Fig 1), carbohydrate intake plus high physical activity level in the second (OR = 0.56; 0.43–0.73) and highest (OR = 0.48; 0.37–0.62) tertiles compared to the lowest tertile were associated with lower T2DM risks. Likewise, among European Americans, high physical activity level with protein intake in the second tertile (OR = 0.45; 0.34-.059) and lowest (OR = 0.35; 0.27–0.46) tertile compared to the highest tertile were associated with lower T2DM risk. Similarly for African Americans (Fig 2), we observed significant T2DM risks for carbohydrate intake in the highest tertile (OR = 0.59; 0.39–0.88) and for protein intake together with high physical activity level in the second (OR = 0.50; 0.34–0.75) and lowest (OR = 0.36; 0.23–0.55) tertiles compared to the highest tertile. When we investigated the reverse (Fig 1), we observed in European Americans that high physical activity decreased the high risks in low carbohydrate and high protein intake as evident in Table 3. We observed a lower T2DM risk in the second tertile (OR = 0.70; 0.53–0.91) compared to the highest tertile (Fig 1). There was lower T2DM risk for high protein intake with high physical activity level that was significant; but this did not meet Bonferroni correction for multiple testing (p < .0125).

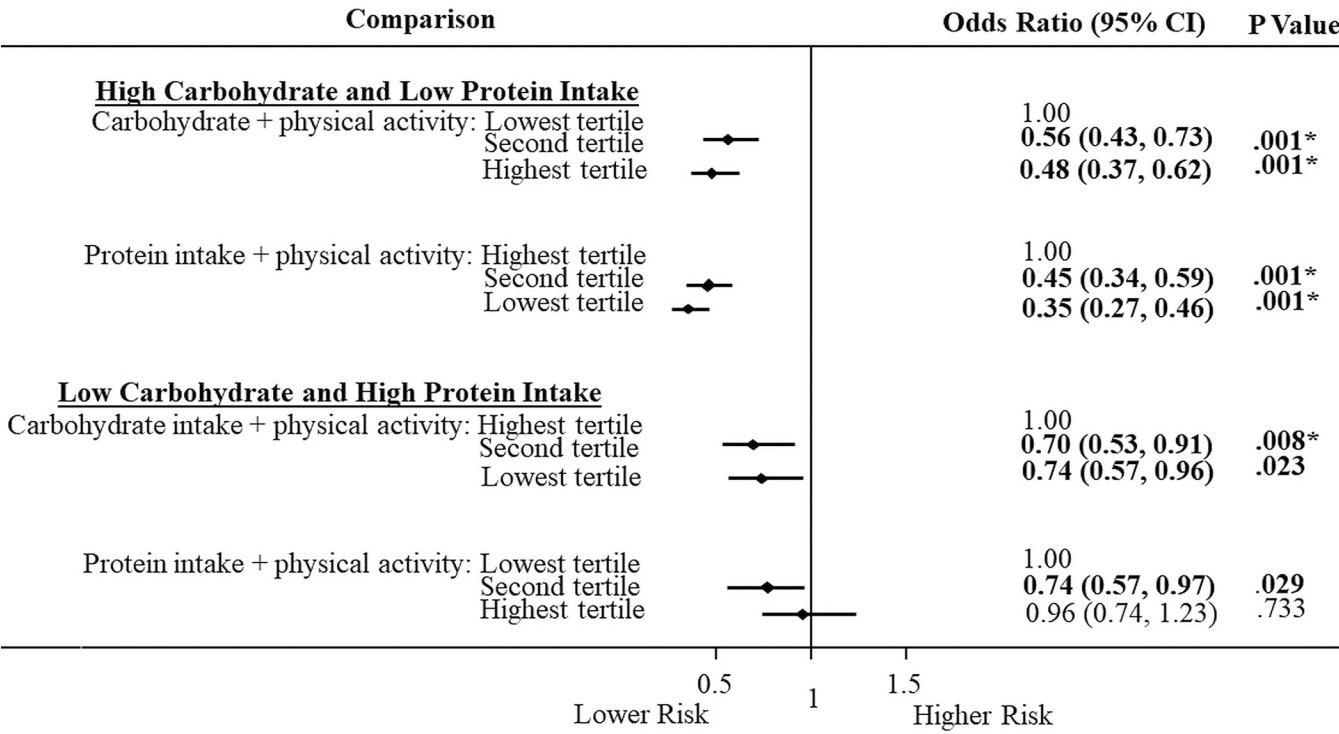

**Fig 1. Association of carbohydrate and protein intake and physical activity with type 2 diabetes in European Americans.**

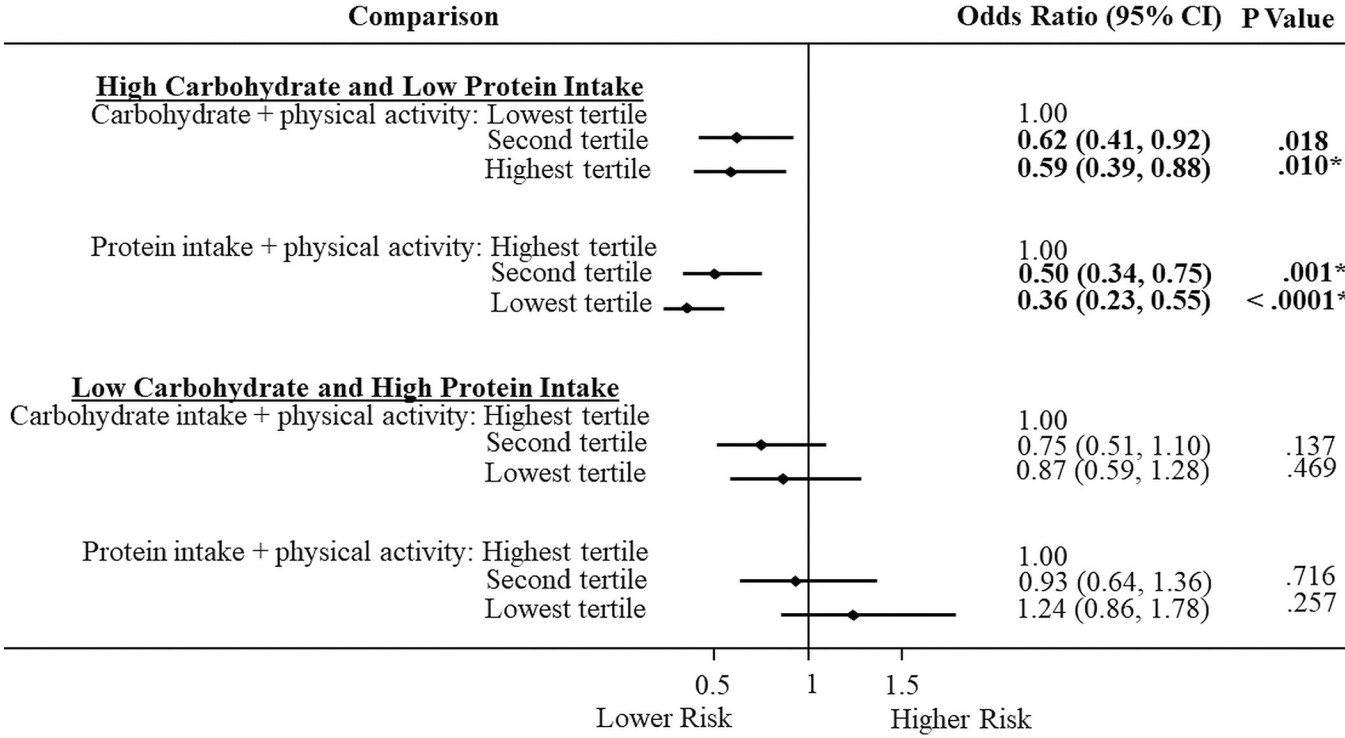

**Fig 2. Association of carbohydrate and protein intake and physical activity with type 2 diabetes in African Americans.**

## Association of the PRS, carbohydrate and protein intake, and physical activity with T2DM

We examined the relationships with the combination of PRS, carbohydrate and protein intake and physical activity level on T2DM (Fig 3). We observed that in models with high or low carbohydrate intake, when combined with the PRS, there were no significant associations with T2DM for both genetic ancestry groups. However, in European Americans, when the PRS was combined with high physical activity level, there were lower risks in the second (OR = 0.56; 0.43–0.74) and highest (OR = 0.61; 0.47–0.80) PRS tertiles. This trend continued when high carbohydrate and high protein intake were added. We observed that T2DM risk decreased when the PRS and carbohydrate intake in the second (OR = 0.53; 0.38–0.74) and highest (OR = 0.49; 0.36–0.68) with high physical activity compared to the PRS and carbohydrate in the lowest tertile with low physical activity. When we combined the PRS with protein intake, there were lower T2DM risks in the highest tertiles compared to the lowest tertiles (OR = 0.63; 0.48–0.83). Likewise, the PRS, low protein intake and high physical activity level in the second (OR = 0.42; 0.30–0.59) and lowest (OR = 0.36; 0.26–0.51) tertiles compared to their lowest

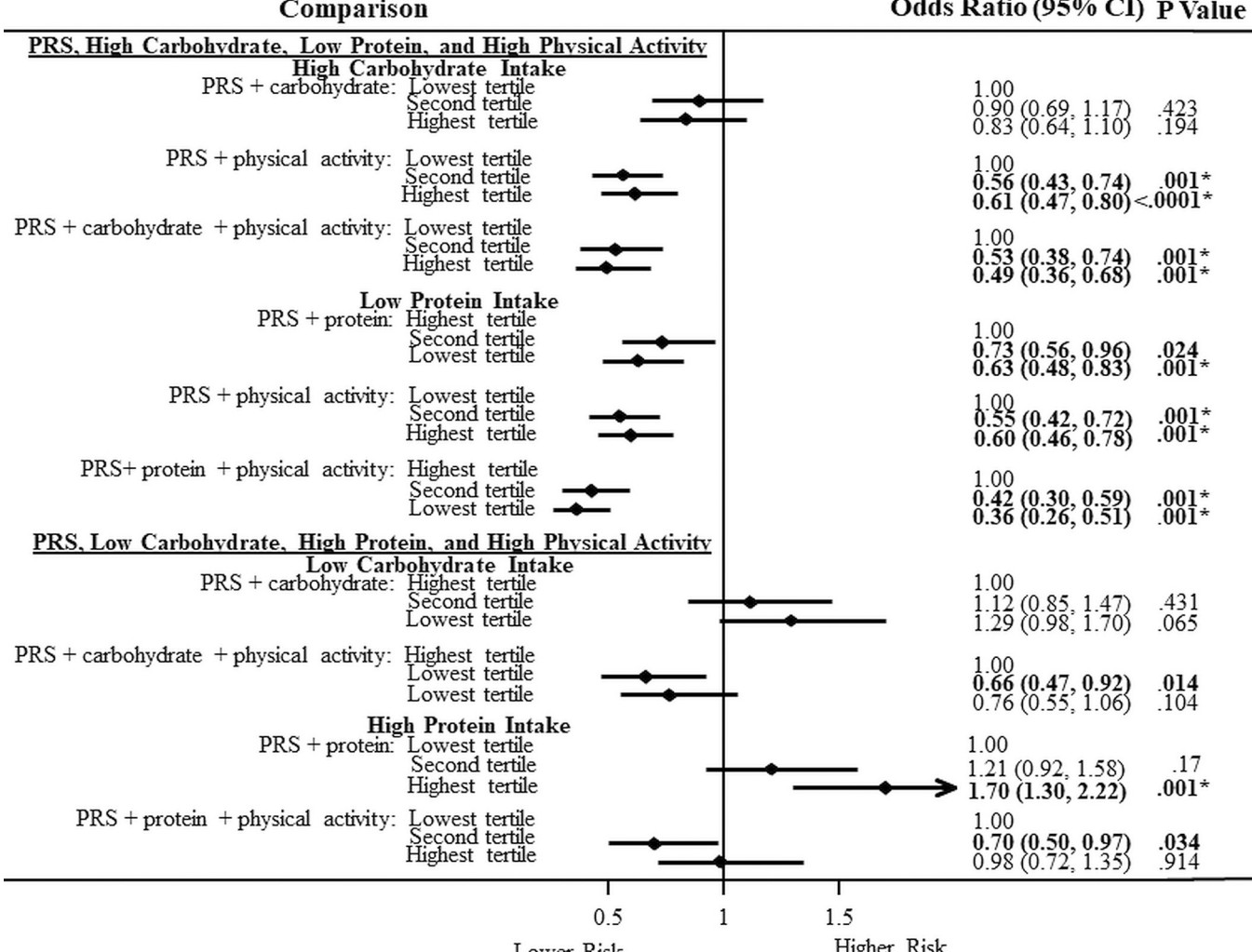

**Fig 3. Association of a polygenic risk score and carbohydrate and protein intake with physical activity on type 2 diabetes in European Americans.**

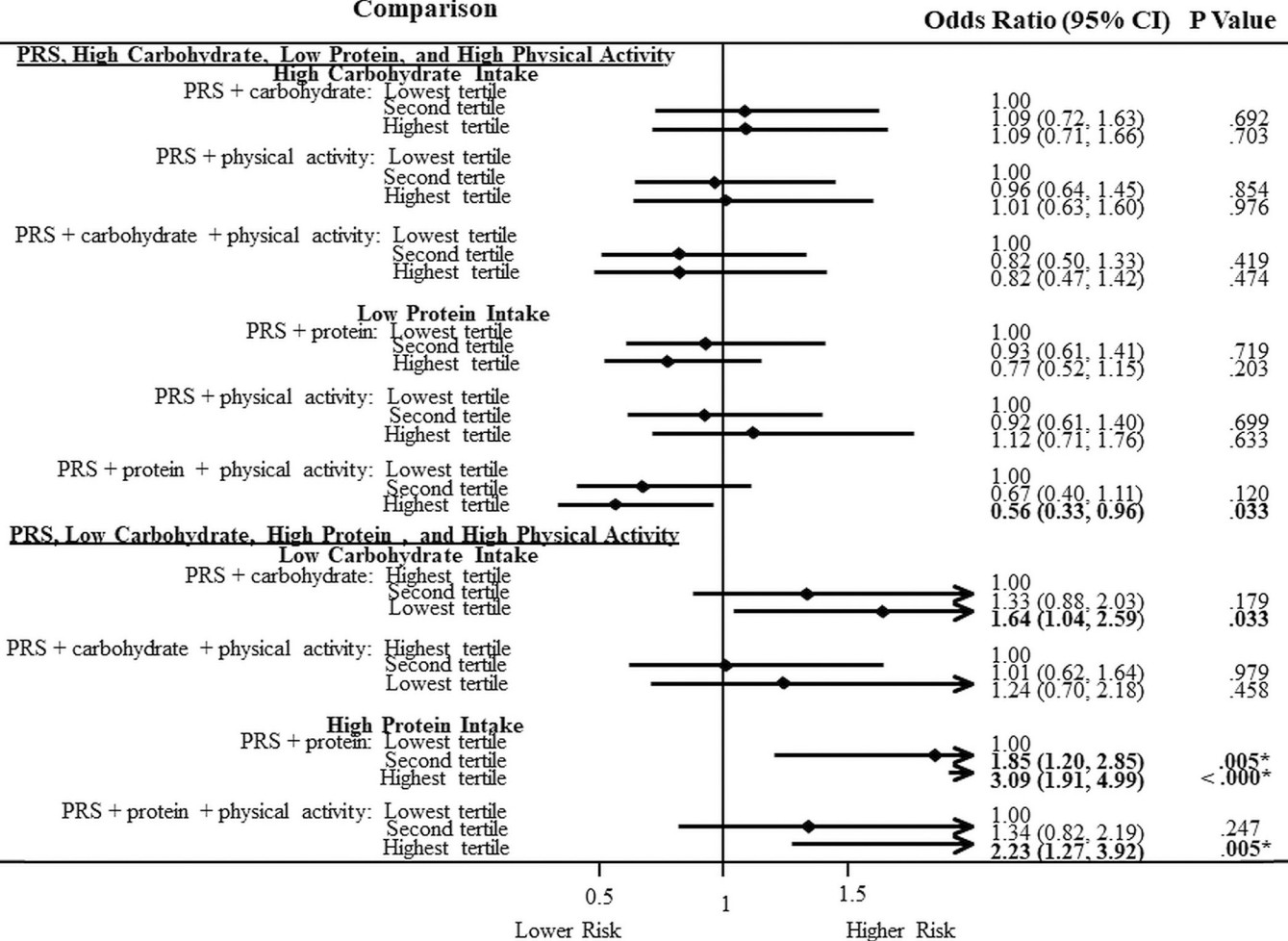

**Fig 4. Association of a polygenic risk score and carbohydrate and protein intake with physical activity on type 2 diabetes in African Americans.**

tertiles and low physical activity were associated with lower T2DM risks. When we looked at the reverse, we saw higher T2DM risks for the PRS with protein intake in the highest tertiles compared to their lowest tertiles (OR = 1.70; 1.30–2.22).

However, for African Americans (Fig 4), in the combination of the risk-raising PRS-high protein diet, we saw a doubling of risk in both the second tertile (OR = 1.85; 1.20–2.85) and in the highest tertile (OR = 3.09; 1.91–4.99) compared to the lowest tertile. The remarkable decreased risk effect of high physical activity was further observed with the risk-raising PRS and high protein diet, by decreasing the association in the highest tertile (OR = 2.23; 1.27–3.92) by a risk difference of 0.86 or by 28.5%.

## Association of carbohydrate and protein intake with T2DM stratified by PRS tertiles, adjusted for all covariates

S4 and S5 Tables show the association of carbohydrate and protein intake stratified by PRS tertiles, adjusted for all covariates in a covariate propensity score. Among European Americans (S4 Table), carbohydrate intake in the highest tertile in combination with the PRS tertile with the highest burden of alleles (highest PRS tertile) was associated with a significant lower risk

effect for T2DM (OR = 0.68; 0.51–0.92). We also observed that in the second and lowest tertiles of protein intake within the lowest PRS and highest PRS tertiles were associated with lower risk for T2DM. When we examined the reverse, we saw a doubling of effects for carbohydrate intake in the second (OR = 1.42;1.06–1.90) and lowest tertile (OR = 1.46;1.09–1.96) within the PRS level with the highest burden of alleles (highest PRS). Among European Americans, a high protein intake in the lowest PRS tertile (OR = 1.83;1.22–2.74) and highest PRS tertile (OR = 1.79;1.34–2.40) were associated with higher risks for T2DM.

Among African Americans, the picture was not as clear (S5 Table). We observed significant risks mainly in the second PRS tertile. When we examined high carbohydrate intake, lower risks were observed in the second PRS tertile, in the second carbohydrate tertile (OR = 0.52; 0.30–0.90) and in the highest carbohydrate tertile (OR = 0.54; 0.31–0.95); but these estimates did not meet Bonferroni cutoff p < .0125. We observed higher risk in the lowest tertile of protein intake (OR = 1.93;1.15–3.21) that barely meet Bonferroni cutoff of p < .0125. When we examined the reverse associations, there was a higher risk for T2DM with a high protein intake in lowest PRS tertile (OR = 2.16; 1.18–3.94) and in the second PRS tertile (OR = 2.65; 1.46–4.81).

## Mediation analysis of the PRS, carbohydrate and protein intake, and physical activity with T2DM

Mediation analysis examines the direct overall effect of the PRS on T2DM and how carbohydrate or protein intake affect this relationship indirectly (Fig 5). We explored whether there was causal mediation using the counterfactual approach described by VanderWeele [31]. The counterfactual situation involves different or absolute measures and relative risk measures where the treated is compared to the untreated (counterfactual) (pg. 49) [36]. In this approach, we examined the mechanism of the PRS through carbohydrate or protein intake on T2DM (Table 4). Among African Americans, we found that protein intake in the highest tertile had causal mediational effects with the PRS in the highest tertile on T2DM. The proportion of

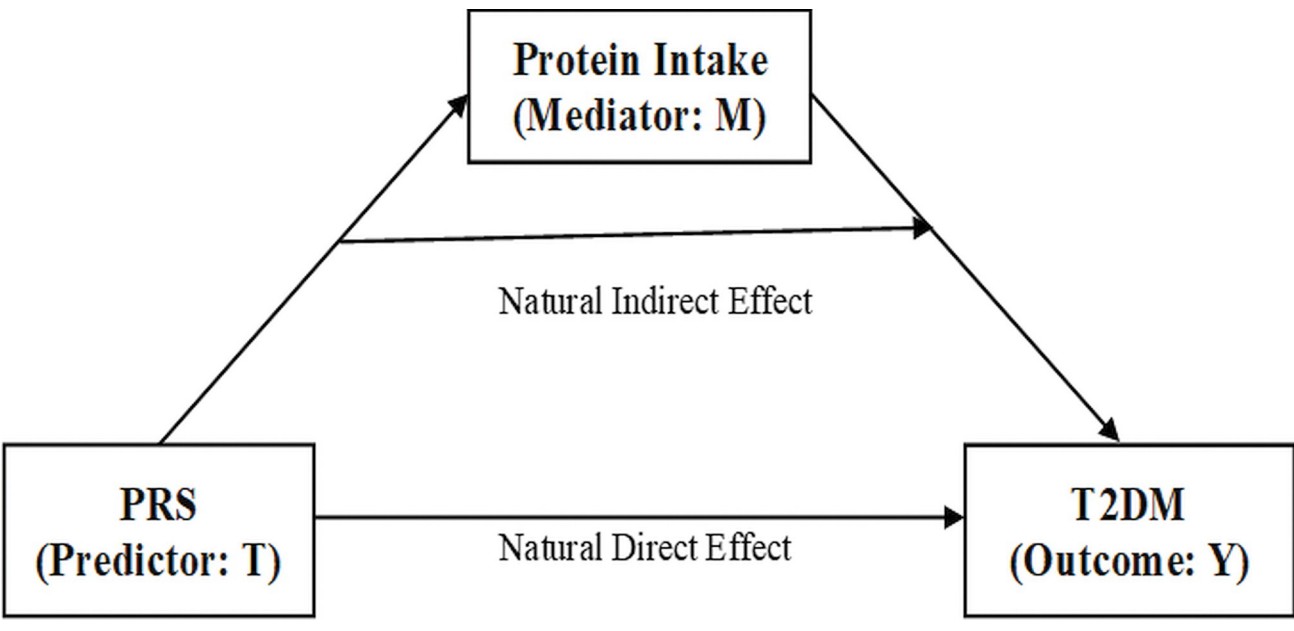

**Fig 5. Causal mediation model.** A path model for protein intake in African Americans.

**Table 4. Causal mediation of PRS with carbohydrate and protein intake with type 2 diabetes.**

| | Odds Ratio(95% Confidence Interval) | | |
|---|---|---|---|
| | Natural Direct Effect | Natural Indirect Effect | Total Causal Effect |
| | **European Americans** | | |
| **PRS & Carbohydrate Intake** | | | |
| Lowest PRS tertile & lowest carbohydrate tertile | 1.00(ref) | 1.00(ref) | 1.00(ref) |
| Second PRS tertile & second carbohydrate tertile | 0.89(0.71–1.13) | 1.00(1.00–1.01) | 0.89(0.71–1.13) |
| Highest PRS tertile & highest carbohydrate tertile | 0.97(0.75–1.25) | 1.00(0.98–1.01) | 0.96(0.75–1.25) |
| **PRS & Protein Intake** | | | |
| Lowest PRS tertile & lowest protein tertile | 1.00(ref) | 1.00(ref) | 1.00(ref) |
| Second PRS tertile & second protein tertile | 1.03(0.81–1.33) | 0.99(0.98–1.00) | 1.03(0.80–1.32) |
| Highest PRS tertile & highest protein tertile | 1.00(0.78–1.30) | 1.02(0.99–1.05) | 1.02(0.79–1.33) |
| | **African Americans** | | |
| **PRS & Carbohydrate Intake** | | | |
| Lowest PRS tertile & lowest carbohydrate tertile | 1.00(ref) | 1.00(ref) | 1.00(ref) |
| Second PRS tertile & second carbohydrate tertile | **1.62(1.07–2.49)** | 0.97(0.91–1.00) | **1.58(1.04–2.41)** |
| Highest PRS tertile & highest carbohydrate tertile | **1.51(1.04–2.19)** | 1.00(0.97–1.01) | **1.51(1.04–2.19)** |
| **PRS & Protein Intake** | | | |
| Lowest PRS tertile & lowest protein tertile | 1.00(ref) | 1.00(ref) | 1.00(ref) |
| Second PRS tertile & second protein tertile | 1.04(0.66–1.62) | 1.02(0.99–1.08) | 1.05(0.67–1.64) |
| Highest PRS tertile & highest protein tertile | **1.74(1.21–2.55)£** | **0.89(0.79–0.96)£** | **1.55(1.08–2.25)*** |

**Abbreviations:** PRS, polygenic risk score.

Causal mediational analysis performed for the PRS and carbohydrate and protein intake with type 2 diabetes to derive odds ratios and 95% confidence intervals. See Fig 1 for more details.

**Bold** indicates p values that are statistically significant at p < .05.

Bonferroni cut-off, *p < .0125.

£, p = .006

African Americans: Proportion protein mediated in highest PRS tertile & highest protein tertile = natural indirect effect/total causal effect = (0.89/1.55)x100 = 55.42%

protein mediated was 55.42%. We next explored whether high physical activity had any effect on the causal mediational relationship for the PRS with carbohydrate or protein intake on T2DM. We did not find any causal mediational effects in these analyses.

## Association of the PRS and metabolic factors with T2DM

We examined metabolic factors in association with the metabolic syndrome criteria [37], dichotomized as high values vs. normal values for waist circumference (men:> 102 cm, women: > 88 cm), elevated systolic blood pressure (≥ 120 mmHg) and blood pressure medications (yes/no) based on the updated blood pressure guidelines [38], low HDL cholesterol (men:< 40 mg/dL, women: < 50 mg/dL), and high triglycerides (≥ 150 mg/dL) [39–41] (S6 Table). Our results showed that as the PRS advanced from the lowest burden of deleterious alleles (lowest tertile) to the highest burden of deleterious alleles (highest tertile), most p values for metabolic risk factors decreased in their association with T2DM. In some metabolic risk factors, the magnitude of effects became greater and more significant. For example, in both genetic ancestry groups, BMI (European Americans: OR = 1.20; p = .080 to OR = 1.32; p = .001 and African Americans: OR = 2.01; P = .043 to OR = 1.86; p = .012) and triglycerides (European Americans: OR = 1.20; p = .080 to OR = 1.32; p = < .0001 and African Americans: OR = 2.95; p = < .0001 to OR = 3.07; p < .0001) generally tended to increase in magnitude and significance across lowest to highest PRS tertiles. However, waist circumference

(OR = 2.06; p = < .0001 to OR = 1.37; p = .031); and hypertensive medications (OR = 3.36; p = < .0001 to OR = 2.10; p = < .0001) decreased in magnitude and in some p value significance in European Americans. This PRS-metabolic pattern was seen more in European Americans compared to African Americans (S5 Table).

## Pathway analysis of SNPs in the PRS mapped to genes

We used snpXplorer to map the SNPs to their respective genes by ancestry [42]. SnpXplorer is a web-based application to explore human SNP-associations and annotate SNP-sets. We then took our gene list by ancestry and analyze the genes by pathways in the Enrichr program. Enrichr uses a set of Entrez gene symbols as input. Fig 6 presents visualizations using the bar graphs. The length of the bar represents the significance of gene-set. The brighter the color, the more significant that term [43]. In the bar graphs, within the top (most significant by p value) 10 pathways for European Americans, we observed pathways for insulin/IGF, mannose metabolism, Alzheimer's disease, and T-cell activation. For African Americans, we observed, within the 10 top pathways, ketogenesis and ketolysis, energy metabolism, MTHFR deficiency, and one carbon metabolism that involve energy and glucose metabolism.

## Discussion

In this study, we found that the PRS with a high burden of deleterious alleles were associated with higher risks for T2DM in both European Americans and African Americans. In general, T2DM risks remained decreased when the PRS was combined with high carbohydrate and low protein intake, even after adjusting for all covariates. In African Americans, the notable effects of high physical activity level were able to decrease the deleterious risks of the PRS-high protein diet combination on T2DM by 28.5%. In addition, among African Americans, protein intake in the highest tertile had mediational effects with a high burden of risk alleles in the highest PRS tertile and mediated 55% of the effect on T2DM. There were differences in top

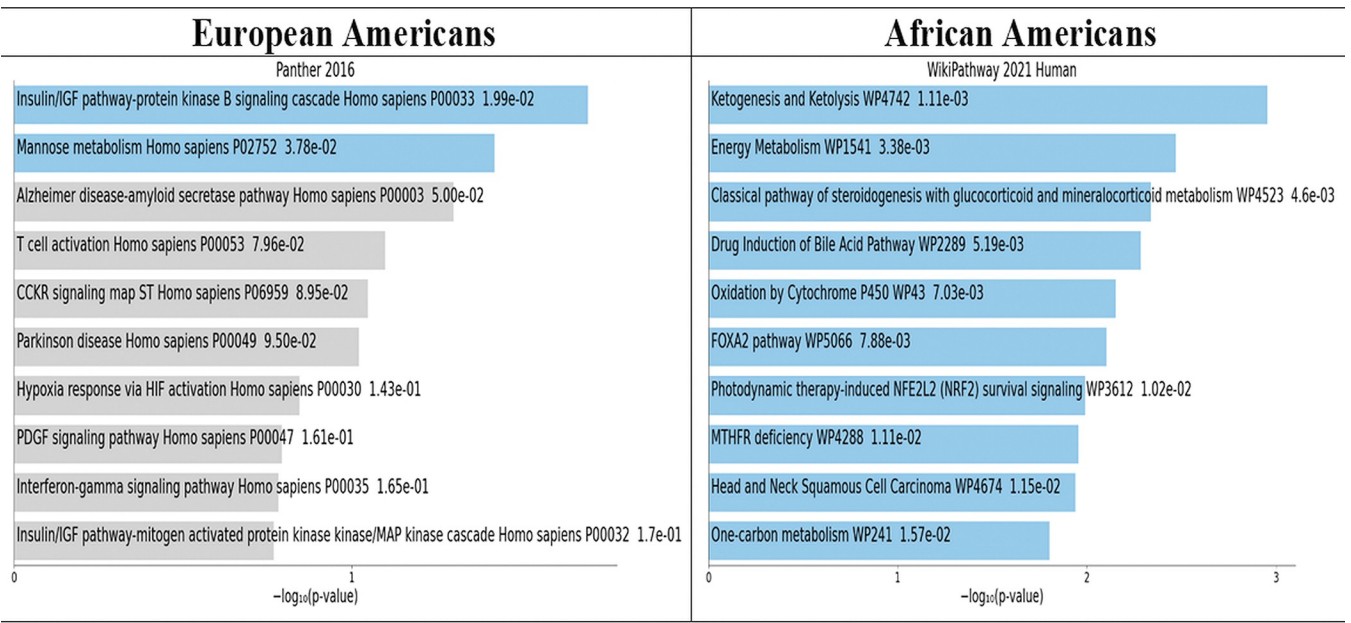

**Fig 6. Pathways of top genes associated with type 2 diabetes by genetic ancestry.**

pathways by genetic ancestry groups. Insulin/IGF and ketogenesis/ketolysis pathways were influenced by genes in European Americans and African Americans, respectively.

In other studies, Liu et al. [6], developed a T2DM PRS that had prediction performance at 90% and was able to identify 7% to 30% of the population who had a 5-fold to 7-fold increased risk for T2DM. Similar to Lee et al. [12] who investigated interaction in a BMI-PRS and caloric intake, we did not find interaction with our T2DM PRS with carbohydrate or protein intake in both ancestries. However, like Krapohl et al. [8] who used the joint effects of SNPs interaction in their PRS, we found that our joint effects functional risk-raising PRS was significantly associated with T2DM. Similar to Kurniansyah et al. [7] who found that their hypertension PRS was associated with T2DM, coronary artery disease, ischemic stroke, and kidney disease, we found that metabolic risk factors within the metabolic syndrome criteria (e.g. waist circumference, systolic blood pressure, HDL cholesterol, etc.) were associated with our risk-raising PRS and that the magnitude of effects and p values generally became stronger in some metabolic risk factors as the PRS mutational burden increased across tertiles, particularly in European Americans.

We found top pathways (most significant in p value) that were associated with genes in the PRSs that have implications with dietary intake. Among European Americans, insulin/IGF pathway-protein kinase B signaling is associated with aging, but dietary restriction can reverse this process [44]. High expression of mannose-lectin binding is associated with vascular complications in diabetes [45]. Because of the acidic nature of the ketogenic diet, overproduction of ketones is a concern. Patients with T2DM and Alzheimer's disease, for example, should consider proceeding with caution because of these harmful effects on different body organs such as the liver [46].

Among African Americans, ketogenesis resulting from intermittent fasting may help with blood glucose control and energy metabolism that is beneficial for T2DM control [47]. Pathways such as insulin/IGF and ketogenesis/ketolysis pathways may be activated by intermittent fasting and could be considered for better T2DM control and T2DM remission [47–50]. Furthermore, habitual physical activity can increase levels of IGF and decrease insulin resistance associated with T2DM [51]. MTHFR has been shown to be associated with T2DM and cardiometabolic diseases. However, medications used to treat T2DM, such as metformin, are likely to deplete the nutrients (B6, B12, folate and CoQ10) known to manage MTHFR [52, 53]. One carbon metabolism is associated with metabolic syndrome [50] which is a precursor to T2DM [54].

The mean carbohydrate intake in the highest tertile in European Americans and African Americans were 58% and 59%, respectively; and for protein intake it was 22% in both ancestries. Some of the data from these studies were collected over 3 decades ago. For example, baseline data from ARIC was collected in 1987 and MESA in 2010. Since then, over the past decades, food quality has decreased, and food quantity has increased as evident by the increased consumption of ultra-processed foods [55]. This has been associated with the continual rise in obesity prevalence and a mirroring effect in T2DM [55, 56].

Low carbohydrate and high protein diets for T2DM patients are currently being prescribed by physicians and other health professionals for optimal blood glucose control, weight loss, and reversal of abnormal metabolic parameters [57]. Although some studies show promising results, the consensus in a systematic review and a meta-analysis show otherwise, that there is no difference in effect for low carbohydrate and high protein diets on blood glucose control in the absence of weight loss [58]. Low carbohydrate and high protein diets may not be superior to other dietary patterns, such as the Mediterranean diet or the vegetarian diet for improvement in high blood glucose [57, 59]. Diets that promote more satiety and less energy consumed, contribute to weight loss and better blood glucose control [60]. Conversely, other studies show concern with high protein diets as it relates to exacerbation of chronic kidney

disease in patients with or without this pre-existing condition [61]. Given that we observed a high proportion of African Americans with T2DM and high blood pressure in this study, we caution the use of high protein diets in this population because T2DM and high blood pressure are risk factors for chronic kidney disease. Combined healthy behaviors such as maintaining healthy body weight, consuming a healthy diet, frequent exercise, smoking abstinence and limited alcohol intake have been associated with 80% lower incident T2DM risk [62].

Our study has some limitations. This study is a cross-sectional study, so only one time-point was used to assess associations. In addition, due to the temporal nature of this study, we were not able to establish a causal relationship between carbohydrate or protein intake and T2DM. Furthermore, because the study was cross-sectional, we did not include follow-up information on changes in food intake, T2DM status or changes in metabolic parameters, which may have had an impact on the T2DM outcome. However, our analyses results show that high carbohydrate and low protein intake were associated with lower T2DM risk (after adjusting for covariates including total calories), even when patients had a high burden of high-risk alleles for T2DM. Another limitation that may be questioned by our readers is that our datasets were NHLBI Care studies from dbGaP. Our reason for using this data source is because of the rich database and the presence of the variables we needed for the project (GWAS data, nutrition variables, physical activity variable, etc.).

A major advantage of this study is that we combined data from seven rigorously pheno-typed NHLBI Care studies, that are generalizable to European Americans and African Americans in the USA. These studies investigated risk factors for atherosclerosis and its sequel; and we used data from these studies to utilize variables for our study. Another advantage was that we were able to impute data to increase the number of genetic markers and phenotypic observations to augment sample size. Although our sample size was diminished after splitting the genetic data into train and test samples, we started with a relatively large sample size in European Americans. However, even though the final African Americans sample size was small, we were still able to discover important associations even when corrected for Bonferroni adjustment. Recall bias and information bias were diminished because of the rigorous nutrition study protocols that were established during the administration of food frequency questionnaires and collection of physical activity information. In addition, we decreased bias in food recall by dropping observations in men and women who were above or below the 1-percentile of total caloric intake.

Historically, the majority of PRSs were derived from genetic variants in populations of European ancestry. To increase the number of variants for African American ancestry, we used variants of European ancestry and applied these to our polygenic risk score in African Americans. The lack of very large genetic studies in African Americans is a concern, because larger samples are needed to address health disparities in this population and their higher risk for T2DM. PRSs may be informative at different stages of T2DM disease trajectory to inform better choices for prevention, assist with diagnosis, and aid with better options for disease management. Nevertheless, a PRS-disease association can be modified by changes in diet and lifestyle factors. The PRS-disease risks can become part of prevention programs and interventions. A high burden of PRS alleles can be a motivating factor to urge people to change their diet and physical activity behaviors to decrease their chances of developing a disease. However, there is a need for more randomized clinical trials and intervention studies to investigate different dietary patterns including low carbohydrate and high protein diets, to assess their utility in decreasing risk for cardiometabolic diseases.

In conclusion, even though our PRSs included genes with deleterious mutations, pathways such as insulin/IGF and ketogenesis/ketolysis can be activated by intermittent fasting and moderate physical activity for better T2DM control. Our PRSs increased the risks for T2DM

and may provide greater evidence when integrated with other medical test results to assist in diagnosis and treatment options, and at different stages of disease to obtain a more accurate measure of disease risk [63]. Participants with high carbohydrate and low protein intake showed lower risk in the highest burden of the PRS risk alleles. A high level of physical activity may lower T2DM risk in African Americans, who consume a harmfully high protein diet despite having a high burden of risk alleles.

## Supporting information

**S1 Table. Datasets utilized in our study from dbGaP. Abbreviations:** dbGaP, Database of Genotypes and Phenotypes; ARIC, Atherosclerosis Risk in Communities study [16]; CARDIA, Coronary Artery Risk Development in Young Adults Study [17]; CHS, Cardiovascular Heart Study [18]; FHS, Framingham Heart Study Offspring and GENX 3 studies [19]; MESA, Multi-Ethnic Study of Atherosclerosis Study [20]; WHI, Women's Health Initiative study [21].
(DOCX)

**S2 Table. Form of covariates used from all NHLBI care datasets.**
(DOCX)

**S3 Table. Characteristics of diet forms used by the seven NHLBI care studies. Abbreviations:** NHLBI, National Heart, Lung, and Blood Institute; Care, Candidate Gene Association Resource; ARIC, Atherosclerosis Risk in Communities study [16], CARDIA, Coronary Artery Risk Development in Young Adults Study [17], CHS, Cardiovascular Heart Study [18], FHS, Framingham Heart Study Offspring and GENX 3 studies [19], MESA, Multi-Ethnic Study of Atherosclerosis Study [20], and Women's Health Initiative study (WHI) [21].
(DOCX)

**S4 Table. Association between a polygenic risk score (PRS) and carbohydrate and protein intake with type 2 diabetes in <u>European Americans</u> adjusted for all covariates.** Bold indicates p values that were statistically significant at $p < .05$.*Bonferroni adjustment for multiple testing for dietary patterns ($p = .05/2 = .0125$). Macronutrient intakes were calculated as percent of total caloric intake. Type 2 diabetes status was regressed against each macronutrient using Generalized Estimating Equation model, adjusting for a covariate propensity score composed of age, sex, physical activity, current cigarette smoking status, current drinking status, carbohydrate, protein, and fat intake stratified by PRS tertiles to derive odds ratios and 95% confidence intervals. When a specific nutrient was the focus of interest, it was not included in the covariate propensity score.
(DOCX)

**S5 Table. Association between a polygenic risk score (PRS) and carbohydrate and protein intake with type 2 diabetes in African Americans.** Bold indicates p values that were statistically significant at $p < .05$. *Bonferroni adjustment for multiple testing ($p < .05/2 = < .0125$). Macronutrient intakes were calculated as percent of total caloric intake. Type 2 diabetes status was regressed against each macronutrient using Generalized Estimating Equation model, adjusted by a covariate propensity score composed of age, sex, physical activity, current cigarette smoking status, current drinking status, carbohydrate, protein, and fat intake stratified by each PRS tertile to derive odds ratios and 95% confidence intervals. When a specific nutrient was the focus of interest, it was not included in the covariate propensity score.
(DOCX)

**S6 Table. Association of a polygenic risk score and metabolic factors with type 2 diabetes. Abbreviations:** PRS, polygenic risk score; HDL, high density cholesterol. General estimating

equation model was constructed by regressing type 2 diabetes status (yes/no) against dichotomized values (high vs. normal) for all metabolic factors (waist circumference, body mass index, systolic blood pressure, etc.) to derive odds ratios and 95% confidence intervals. adjusted for a propensity score that consisted of age, sex, physical activity, current smoking, current drinking, and 10 principal components for population stratification.
(DOCX)

## Acknowledgments

The authors would like to thank Dr. Bamidele Tayo, PhD for his guidance in preparing datasets for genotyping imputation, and analysis of the genetic datasets for PRS preparation.

## Author Contributions

**Conceptualization:** Dale S. Hardy.

**Data curation:** Dale S. Hardy.

**Formal analysis:** Dale S. Hardy.

**Funding acquisition:** Dale S. Hardy.

**Investigation:** Dale S. Hardy.

**Methodology:** Dale S. Hardy, Jane T. Garvin, Tesfaye B. Mersha.

**Project administration:** Dale S. Hardy, Jane T. Garvin, Tesfaye B. Mersha.

**Resources:** Dale S. Hardy.

**Software:** Dale S. Hardy.

**Supervision:** Dale S. Hardy, Jane T. Garvin, Tesfaye B. Mersha.

**Validation:** Dale S. Hardy, Jane T. Garvin, Tesfaye B. Mersha.

**Visualization:** Dale S. Hardy, Jane T. Garvin, Tesfaye B. Mersha.

**Writing – original draft:** Dale S. Hardy.

**Writing – review & editing:** Dale S. Hardy, Jane T. Garvin, Tesfaye B. Mersha.

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
