## [Decision Letter · Decision Letter 0]

20 Feb 2023

PONE-D-22-35678Analysis of Ancestry-Specific Polygenic Risk Score and Diet Composition in Type 2 DiabetesPLOS ONE

Dear Dr. Hardy,

Thank you for submitting your manuscript to PLOS ONE. After careful consideration, we feel that it has merit but does not fully meet PLOS ONE’s publication criteria as it currently stands. Therefore, we invite you to submit a revised version of the manuscript that addresses the points raised during the review process. Particularly, the authors need to clarify and discuss any potential bias generated by the data and its interpretation as detailed by Reviewer 2.

We look forward to receiving your revised manuscript.

Kind regards,

Andreas Zirlik, MD

Academic Editor

PLOS ONE

Journal Requirements:

"This work is supported by a K01 grant from the National Heart, Lung, and Blood Institute grants: K01 HL127278 awarded to Dale Hardy, PhD."

"This work is supported by a K01 grant from the National Heart, Lung, and Blood Institute grants: K01 HL127278 awarded to Dale Hardy, PhD."

"NO authors have competing interests"

7. We note that you have included the phrase “data not shown” in your manuscript. Unfortunately, this does not meet our data sharing requirements. PLOS does not permit references to inaccessible data. We require that authors provide all relevant data within the paper, Supporting Information files, or in an acceptable, public repository. Please add a citation to support this phrase or upload the data that corresponds with these findings to a stable repository (such as Figshare or Dryad) and provide and URLs, DOIs, or accession numbers that may be used to access these data. Or, if the data are not a core part of the research being presented in your study, we ask that you remove the phrase that refers to these data.

8. Please include your full ethics statement in the ‘Methods’ section of your manuscript file. In your statement, please include the full name of the IRB or ethics committee who approved or waived your study, as well as whether or not you obtained informed written or verbal consent. If consent was waived for your study, please include this information in your statement as well. 

Additional Editor Comments:

Reviewer 1:

In short summary, this study found that individuals in the highest PRS tertile had higher T2DM risk in European Americans and African Americans. Protective effects were seen with combination of high carbohydrate, low protein intake and low PRS. African Americans benefited from increased physical activity by reduce harm of high protein diet.

Sections:

1. Disclosures:

a. In financial disclosures, the authors mention a grant from the NLHBI to the first author. As this grant (K01) can include salaries and only NLHBI-sponsored studies were accessed, this should also be mentioned as conflict of interest. Otherwise, the authors should explain why this should not be the case.

2. Introduction:

a. The authors should shortly state the clinical importance of their work at the end of the introduction and furthermore in the discussion and conclusion section.

3. Results:

a. Bonferroni adjustments: The authors make a Bonferroni adjustment for both African American and European Americans, i.e. deduce a significance level of .025. I would like the authors to explain why a Bonferroni adjustment was not done for testing between different tertiles (e.g. in Table 3), as this is also a case of multiple testing. However, I am aware that this topic can be interpreted differentially. This would reduce the number of reported significant associations in the tables significantly.

b. Tables 4 and 7 are quite hard to interpret as the information is somewhat ambiguous. The tables should be simplified or restructured to improve readability. Also, instructions of how to interpret the table should be given in the caption. For example, in Table 4, the authors compare tertiles within two groups, one with “High Carbohydrate and Low Protein Intake” and vice versa. Then, in the former group, the authors present data about Physical activity alone twice.

4. Conclusions:

a. “African Americans can benefit from increased physical activity to decrease the harmful effects of a high protein diet despite having a deleterious burden of high-risk alleles.” (This is a literal quote from the abstract).

This study only tested for associations, not causality, which should be stated in the paragraph listing the study’s limitations. The design of the study is a cross-sectional study that aims to examine the relationship between variables at one point in time, rather than to establish cause-and-effect relationships. Thus, for the next iteration of this work, the authors should rephrase this sentence such as “African Americans may benefit…” etc. as they did in the conclusions section in the main text.

b. The authors should elaborate on the clinical significance of this paper, i.e. hypothesis generation for future RCTs or direct clinical implications.

5. General remarks:

a. Length: The article is quite long and very technical. Perhaps, the authors can omit some of the presented text and tables and transfer them into the supplement.

b. Captions: In general, captions should be optimized to give the reader more guidance of how the data has been interpreted.

Misspellings and Errors:

1. Line numbers suddenly stop at line 250 without apparent reason.

2. Line 65: should be "...that affects 37.3M (11.3%) people in the U.S."

3. Line 99: "was associated". Replace with "has been associated".

4. In the last sentence of the main text should be changed from “..harmful high protein diet…” to “ A high level of physical activity may be protective in African Americans with T2DM who consume a harmfully high protein diet…”.

Reviewer 2:

The authors performed a retrospective study of the influence of a polygenic risk score (PRS), diet and physical activity on the occurrence of diabetes mellitus type 2 (T2DM). They showed in an analysis from many genetic studies that a PRS influenced the occurrence of DMT2. However, high carbohydrate and low protein intake were associated with reduced risk of T2DM.

This observational study contradicts the results of prospective interventional studies, probably due to bias: Since more than 15 years, low carbohydrate diet was led to improved glycaemic control in T2DM patients (also in randomized studies) [Nutr Metab (Lond). 2008; 5: 9.]. Therefore, national and international guidelines mainly recommended carbohydrate restriction in T2DM patients. (In the last years, there was a change towards the assessment of the quality of carbohydrates, but this is still a matter of debate). Consequently, all patients with T2DM in the last decades received recommendations to reduce carbohydrate intake and increase protein intake. Patients included in the genetic studies may also have received the same recommendation and therefore either (1) have changed their dietary habits or (2) at least report that they would have changed their dietary habits in interviews. This would have led to this carbohydrate “paradox” in this study. What do the authors think about this problematic bias? I encourage the authors to clearly state in the discussion that this is only a retrospective study with associations. I discourage the use of the word “protective” as the study has no longitudinal data included, therefore no “protective” behavioural factors can be derived from this data. Furthermore, the authors discuss the possibility of “ketogenic diet” with means a total cut of carbohydrates and clearly contradicts the results of this study.

Furthermore, the authors should describe the source data better: Were those cross-sectional or longitudinal data? (this should be already mentioned in the Methods section). How were patients included? Do the patients represent the same characteristics as the general population? Did patients receive compensation?

In general, the manuscript is well written, statistics and references are adequate. The manuscript is generally very long, the results section may be shortened.

Minor comments:

- The first half of the first paragraph at page 14 should be moved to the Methods section.

- The authors write that the ADA currently recommends a high carbohydrate, low protein diet with reference [9]. However, the reference [9] refers to the definition of T2DM, and does not recommend any dietary measure. In fact, this reference does not even contain the words “carbohydrate” or “protein”.

- The main endpoint (occurrence of T2DM) is not clear from the abstract.

- Specific steps and changed variables during data harmonization should be summarized in a supplemental table.

- Was multiple imputation performed for missing data?

Reviewers' comments:

Reviewer's Responses to Questions

**Comments to the Author**

1. Is the manuscript technically sound, and do the data support the conclusions?

Reviewer #1: Yes

Reviewer #2: Yes

2. Has the statistical analysis been performed appropriately and rigorously? 

Reviewer #1: Yes

Reviewer #2: Yes

3. Have the authors made all data underlying the findings in their manuscript fully available?

Reviewer #1: No

Reviewer #2: Yes

4. Is the manuscript presented in an intelligible fashion and written in standard English?

Reviewer #1: Yes

Reviewer #2: Yes

5. Review Comments to the Author

Reviewer #1: In short summary, this study found that individuals in the highest PRS tertile had higher T2DM risk in European Americans and African Americans. Protective effects were seen with combination of high carbohydrate, low protein intake and low PRS. African Americans benefited from increased physical activity by reduce harm of high protein diet.

Sections:

1. Disclosures:

a. In financial disclosures, the authors mention a grant from the NLHBI to the first author. As this grant (K01) can include salaries and only NLHBI-sponsored studies were accessed, this should also be mentioned as conflict of interest. Otherwise, the authors should explain why this should not be the case.

2. Introduction:

a. The authors should shortly state the clinical importance of their work at the end of the introduction and furthermore in the discussion and conclusion section.

3. Results:

a. Bonferroni adjustments: The authors make a Bonferroni adjustment for both African American and European Americans, i.e. deduce a significance level of .025. I would like the authors to explain why a Bonferroni adjustment was not done for testing between different tertiles (e.g. in Table 3), as this is also a case of multiple testing. However, I am aware that this topic can be interpreted differentially. This would reduce the number of reported significant associations in the tables significantly.

b. Tables 4 and 7 are quite hard to interpret as the information is somewhat ambiguous. The tables should be simplified or restructured to improve readability. Also, instructions of how to interpret the table should be given in the caption. For example, in Table 4, the authors compare tertiles within two groups, one with “High Carbohydrate and Low Protein Intake” and vice versa. Then, in the former group, the authors present data about Physical activity alone twice.

4. Conclusions:

a. “African Americans can benefit from increased physical activity to decrease the harmful effects of a high protein diet despite having a deleterious burden of high-risk alleles.” (This is a literal quote from the abstract).

This study only tested for associations, not causality, which should be stated in the paragraph listing the study’s limitations. The design of the study is a cross-sectional study that aims to examine the relationship between variables at one point in time, rather than to establish cause-and-effect relationships. Thus, for the next iteration of this work, the authors should rephrase this sentence such as “African Americans may benefit…” etc. as they did in the conclusions section in the main text.

b. The authors should elaborate on the clinical significance of this paper, i.e. hypothesis generation for future RCTs or direct clinical implications.

5. General remarks:

a. Length: The article is quite long and very technical. Perhaps, the authors can omit some of the presented text and tables and transfer them into the supplement.

b. Captions: In general, captions should be optimized to give the reader more guidance of how the data has been interpreted.

Misspellings and Errors:

1. Line numbers suddenly stop at line 250 without apparent reason.

2. Line 65: should be "...that affects 37.3M (11.3%) people in the U.S."

3. Line 99: "was associated". Replace with "has been associated".

4. In the last sentence of the main text should be changed from “..harmful high protein diet…” to “ A high level of physical activity may be protective in African Americans with T2DM who consume a harmfully high protein diet…”.

Reviewer #2: The authors performed a retrospective study of the influence of a polygenic risk score (PRS), diet and physical activity on the occurrence of diabetes mellitus type 2 (T2DM). They showed in an analysis from many genetic studies that a PRS influenced the occurrence of DMT2. However, high carbohydrate and low protein intake were associated with reduced risk of T2DM.

This observational study contradicts the results of prospective interventional studies, probably due to bias: Since more than 15 years, low carbohydrate diet was led to improved glycaemic control in T2DM patients (also in randomized studies) [Nutr Metab (Lond). 2008; 5: 9.]. Therefore, national and international guidelines mainly recommended carbohydrate restriction in T2DM patients. (In the last years, there was a change towards the assessment of the quality of carbohydrates, but this is still a matter of debate). Consequently, all patients with T2DM in the last decades received recommendations to reduce carbohydrate intake and increase protein intake. Patients included in the genetic studies may also have received the same recommendation and therefore either (1) have changed their dietary habits or (2) at least report that they would have changed their dietary habits in interviews. This would have led to this carbohydrate “paradox” in this study. What do the authors think about this problematic bias? I encourage the authors to clearly state in the discussion that this is only a retrospective study with associations. I discourage the use of the word “protective” as the study has no longitudinal data included, therefore no “protective” behavioural factors can be derived from this data. Furthermore, the authors discuss the possibility of “ketogenic diet” with means a total cut of carbohydrates and clearly contradicts the results of this study.

Furthermore, the authors should describe the source data better: Were those cross-sectional or longitudinal data? (this should be already mentioned in the Methods section). How were patients included? Do the patients represent the same characteristics as the general population? Did patients receive compensation?

In general, the manuscript is well written, statistics and references are adequate. The manuscript is generally very long, the results section may be shortened.

Minor comments:

- The first half of the first paragraph at page 14 should be moved to the Methods section.

- The authors write that the ADA currently recommends a high carbohydrate, low protein diet with reference [9]. However, the reference [9] refers to the definition of T2DM, and does not recommend any dietary measure. In fact, this reference does not even contain the words “carbohydrate” or “protein”.

- The main endpoint (occurrence of T2DM) is not clear from the abstract.

- Specific steps and changed variables during data harmonization should be summarized in a supplemental table.

- Was multiple imputation performed for missing data?

6. PLOS authors have the option to publish the peer review history of their article (what does this mean?). If published, this will include your full peer review and any attached files.

Reviewer #1: No

Reviewer #2: No

---

## [Author Response · Author response to Decision Letter 0]

7 Apr 2023

We uploaded the response to the reviewers. We attempted our best as possible to answer the comments in detail. Please see this document in the uploaded package.

---

## [Decision Letter · Decision Letter 1]

27 Apr 2023

PONE-D-22-35678R1Analysis of Ancestry-Specific Polygenic Risk Score and Diet Composition in Type 2 DiabetesPLOS ONE

Dear Dr. Hardy,

Thank you for submitting your manuscript to PLOS ONE. After careful consideration, we feel that it has merit but does not fully meet PLOS ONE’s publication criteria as it currently stands. Therefore, we invite you to submit a revised version of the manuscript that addresses the points raised during the review process.

We look forward to receiving your revised manuscript.

Kind regards,

Andreas Zirlik, MD

Academic Editor

PLOS ONE

Journal Requirements:

Reviewers' comments:

Reviewer's Responses to Questions

**Comments to the Author**

1. If the authors have adequately addressed your comments raised in a previous round of review and you feel that this manuscript is now acceptable for publication, you may indicate that here to bypass the “Comments to the Author” section, enter your conflict of interest statement in the “Confidential to Editor” section, and submit your "Accept" recommendation.

Reviewer #1: All comments have been addressed

Reviewer #2: (No Response)

2. Is the manuscript technically sound, and do the data support the conclusions?

Reviewer #1: Yes

Reviewer #2: Yes

3. Has the statistical analysis been performed appropriately and rigorously? 

Reviewer #1: Yes

Reviewer #2: Yes

4. Have the authors made all data underlying the findings in their manuscript fully available?

Reviewer #1: Yes

Reviewer #2: Yes

5. Is the manuscript presented in an intelligible fashion and written in standard English?

Reviewer #1: Yes

Reviewer #2: Yes

6. Review Comments to the Author

Reviewer #1: This retrospective study found that individuals within a higher PRS percentile had increased risk of developing T2DM. Furthermore, there was an association between a high protein low carbohydrate diet and risk of developing T2DM.

In my opinion, my posed questions have been sufficiently addressed. Especially the newly added captions to the figures enhance readability.

Reviewer #2: The authors made substantial improvements to the manuscript. However, I believe that the manuscript can still be optimized:

- As already mentioned in the first review, the main outcome has to be reported in the abstract. I therefore recommend to add this information to the Methods section of the manuscript.

- There is still missing information about data imputation in the manuscript (imputation by mean? Or median? Multiple imputation?)

7. PLOS authors have the option to publish the peer review history of their article (what does this mean?). If published, this will include your full peer review and any attached files.

Reviewer #1: No

Reviewer #2: No

---

## [Author Response · Author response to Decision Letter 1]

27 Apr 2023

Reviewers’ Comments:

6. Review Comments to the Author

Comments Responses

Reviewer #1: This retrospective study found that individuals within a higher PRS percentile had increased risk of developing T2DM. Furthermore, there was an association between a high protein low carbohydrate diet and risk of developing T2DM.

In my opinion, my posed questions have been sufficiently addressed. Especially the newly added captions to the figures enhance readability.

 Thank you for your comment. Your contribution was instrumental in improving the paper to make it better. 

Reviewer #2: The authors made substantial improvements to the manuscript. However, I believe that the manuscript can still be optimized:

- As already mentioned in the first review, the main outcome has to be reported in the abstract. I therefore recommend to add this information to the Methods section of the manuscript.

 Thank you for pointing out this oversight. We have added a sentence to address the main outcome as type 2 diabetes in the METHODS section of the ABSTRACT. 

- There is still missing information about data imputation in the manuscript (imputation by mean? Or median? Multiple imputation?)

 Thank you for your recommendation. We have added the missing information about data imputation under the sub-section Statistical analysis on page 8, lines 176-179.

---

## [Editor Report · Decision Letter 2]

3 May 2023

Analysis of Ancestry-Specific Polygenic Risk Score and Diet Composition in Type 2 Diabetes

PONE-D-22-35678R2

Dear Dr. Hardy,

We’re pleased to inform you that your manuscript has been judged scientifically suitable for publication and will be formally accepted for publication once it meets all outstanding technical requirements.

Kind regards,

Andreas Zirlik, MD

Academic Editor

PLOS ONE
---

## [Editor Report · Acceptance letter]

15 May 2023

PONE-D-22-35678R2 

Analysis of Ancestry-Specific Polygenic Risk Score and Diet Composition in Type 2 Diabetes 

Dear Dr. Hardy:

I'm pleased to inform you that your manuscript has been deemed suitable for publication in PLOS ONE. Congratulations! Your manuscript is now with our production department. 

Kind regards, 

on behalf of

Univ. Prof. Dr. Andreas Zirlik 

Academic Editor

PLOS ONE